# Development and validation of a nomogram for predicting cefoperazone/sulbactam-induced hypoprothrombinaemia in Hospitalized adult patients

Hehe Bai[ID][1], Huan Li[2], Xiaojing Nie[1], Yanqin Yao[3], Xiaonian Han[1], Jinping Wang[1], Lirong Peng[ID][1] *

1 Department of Pharmacy, Xi' an Central Hospital, Xi'an, Shaanxi, China, 2 School of Pharmaceutical Sciences, Tsinghua University, Beijing, China, 3 Department of Pharmacy, The Third Affiliated Hospital of Xi 'an Medical University, Xi'an, Shaanxi, China

* 519925564@qq.com

**Data Availability Statement:** All relevant data are within the paper and its Supporting Information files.

## Abstract

Cefoperazone/sulbactam-induced hypoprothrombinaemia is associated with longer hospital stays and increased risk of death. The aim of this study was to develop and validate a nomogram for predicting the occurrence of cefoperazone/sulbactam-induced hypoprothrombinaemia in hospitalized adult patients. This retrospective cohort study involved hospitalized adult patients at Xi'an Central Hospital from January 2020 to December 2022 based on the Chinese pharmacovigilance system developed and established by the Adverse Drug Reaction Monitoring Center in China. Independent predictors of cefoperazone/sulbactam-induced hypoprothrombinaemia were obtained using multivariate logistic regression and were used to develop and establish the nomogram. According to the same standard, the clinical data of hospitalized patients using cefoperazone/sulbactam at the Third Affiliated Hospital of Xi'an Medical University from January 1, 2023 to June 30, 2023 were collected as the external validation group. The 893 hospitalized patients included 95 who were diagnosed with cefoperazone/sulbactam-induced hypoprothrombinaemia. Our study enrolled 610 patients: 427 in the training group and 183 in the internal validation group. The independent predictors of cefoperazone/sulbactam-induced hypoprothrombinaemia were surgery (odds ratio [OR] = 5.279, 95% confidence interval [CI] = 2.597–10.729), baseline platelet count $\leq$50×10$^9$/L (OR = 2.492, 95% CI = 1.110–5.593), baseline hepatic dysfunction (OR = 12.362, 95% CI = 3.277–46.635), cumulative defined daily doses (OR = 1.162, 95% CI = 1.162–1.221) and nutritional risk (OR = 16.973, 95% CI = 7.339–39.254). The areas under the curve (AUC) of the receiver operating characteristic for the training and internal validation groups were 0.909 (95% CI = 0.875–0.943) and 0.888 (95% CI = 0.832–0.944), respectively. The Hosmer-Lemeshow tests yielded $p$ = 0.475 and $p$ = 0.742 for the training and internal validation groups, respectively, confirming the goodness of fit of the nomogram model. In the external validation group (n = 221), the nomogram was equally robust in cefoperazone/sulbactam-induced hypoprothrombinaemia (AUC = 0.837, 95%CI = 0.736–0.938). The nomogram model constructed in this study had good predictive performance

**Funding:** This work was supported by the Xi'an Science and Technology Program (No.22YXYJ0015) and the Scientific Research Foundation of Xi 'an Central Hospital (No.2022YB06). The funders had no role in study design, data collection and analysis, decision to publish, or preparation of the manuscript.

**Competing interests:** The authors have declared that no competing interests exist.

and extrapolation, which can help clinicians to identify patients at high risk of cefoperazone/sulbactam-induced hypoprothrombinaemia early. This will be useful in preventing the occurrence of cefoperazone/sulbactam-induced hypoprothrombinaemia and allowing timely intervention measures to be performed.

## Introduction

Cefoperazone/sulbactam is a combination of the third-generation cephalosporin antibiotic cefoperazone and the β-lactamase inhibitor sulbactam at certain proportions. Sulbactam can protect cefoperazone from hydrolysis via β-lactamase to expand the antibacterial spectrum of cefoperazone and enhance antibacterial activity [1, 2]. It is widely used in clinical practice for moderate to severe infections caused by Gram-positive cocci [3], Gram-negative bacilli [4] and anaerobic bacteri [5]. The main safety concern of cefoperazone/sulbactam is the occurrence of hypoprothrombinaemia. There is currently a lack of accurate incidence data on cefoperazone/sulbactam-induced hypoprothrombinaemia among all hospitalized patients. In the previous literature, the incidence of cefoperazone/sulbactam-induced hypoprothrombinaemia in hospitalized patients ranged from 4% to 68% depending on the definition of this adverse event and the study population [6]. Clinical manifestations include haematuria and subcutaneous, gastrointestinal and cerebral bleeding, which are associated with increased treatment difficulty and longer hospital stay [7, 8], and can sometimes be fatal [9]. Cefoperazone can cause longer prothrombin time (PT), coagulation disorders and bleeding by interfering with vitamin K metabolism [10]. Article 13 of the State Drug Administration of China established in 2019 therefore requires the revision of the drug instructions for cefoperazone, adding "thrombocytopenia, hypoprothrombinaemia, coagulation disorders and bleeding" under the adverse-reactions item, and adding warnings about hypoprothrombinaemia and bleeding risk under the precautions item [11].

Hypoprothrombinaemia seems to be a more common adverse reaction during treatment with cefoperazone/sulbactam than is generally acknowledged. However, prevention is the key to avoid the occurrence of life-threatening bleeding events [7]. Early detection of patients with a risk of hypoprothrombinaemia caused by cefoperazone/sulbactam will facilitate the effective utilization of medical resources, performing timely intervention measures and ensuring patient safety. Although increased values of the parameters for coagulation is the cornerstone of the diagnosis of hypoprothrombinaemia caused by cefoperazone/sulbactam, early recognition is hindered by many factors and is less accurate and timely [12]. Hypoprothrombinaemia biomarkers, such as prothrombin activity and vitamin-K-dependent factors II, VII, IX and X, may improve the accuracy of risk assessment, but these tests are still far from widespread clinical application. The early identification of antibacterial-associated hypoprothrombinaemia is currently strongly dependent on the laboratory testing frequency. However, patients were found to not be routinely screened for coagulation parameters during cefoperazone/sulbactam use in clinical practice. Even when hypoprothrombinaemia is identified, medical staff may prioritize disease factors over drug-induced ones. Acidosis [13] or hypocalcaemia [14] may also interfere with coagulation function assessments and hinder early identification of risk in some patients. It is therefore particularly important to develop a convenient, accurate and efficient model for predicting cefoperazone/sulbactam-induced hypoprothrombinaemia. Nomograms are currently widely utilized to predict the occurrence, recurrence and prognosis of diseases due to their visual nature and being easy to understand [15, 16]. However, nomograms are mostly used to optimize the administration scheme of antibiotics in specific populations [17,

18], and are rarely used to predict the risk of adverse drug reactions (ADRs) to antibiotics in the general hospital population.

Therefore, a retrospective cohort study was conducted to explore the clinical characteristics of patients with cefoperazone/sulbactam-induced hypoprothrombinaemia based on the Chinese hospital pharmacovigilance system (CHPS) developed and established by the Adverse Drug Reaction Monitoring Center of China, and a simple and easy-to-use nomogram was constructed to help clinicians more quickly and accurately identify patients with a potential risk of cefoperazone/sulbactam-induced hypoprothrombinaemia.

## Methods

### Study design and patient collection

This was a retrospective cohort study of a group of hospitalized patients aged 18 years and older, commencing in March 2023. Patients who were treated using cefoperazone/sulbactam at Xi'an Central Hospital between 1 January 2020 and 31 December 2022 were enrolled. All patient data including medical records and examination information were extracted from the hospital information system (HIS). The HIS can sort out and integrate the data from laboratory information system (LIS), picture archiving and communication system (PACS) and radiology information system (RIS). The CHPS can correspond with the HIS database to obtain patient information. The CHPS can perform intelligent searches and actively monitor hospital prescription events based on the knowledge base of ADRs and search engine technology. The CHPS can initially judge whether the patient has cefoperazone/sulbactam-induced hypoprothrombinaemia and give warning signals once the monitoring indicators are triggered. The cases that required early warnings were actively captured by developing a monitoring plan for cefoperazone/sulbactam. These cases were subsequently independently reviewed by two clinical pharmacists to confirm the monitoring results. If the results were inconsistent, the cases were transferred to clinical experts for the final judgement of whether cefoperazone/sulbactam-induced hypoprothrombinaemia had occurred. The study was approved by the Ethics Committee of Xi'an Central Hospital (No. LW–2023–014).

We defined cefoperazone/sulbactam-induced hypoprothrombinaemia as an increase of 25% in the baseline value of either PT or activated partial thromboplastin time (APTT) after cefoperazone/sulbactam was taken. The study inclusion criteria were as follows: (1) age ≥18 years and (2) the course of intravenous cefoperazone/sulbactam administration lasting ≥24 hours. The exclusion criteria were as follows: (1) baseline coagulation parameters exceeded 25% of the upper limit of normal; (2) absent baseline or follow-up coagulation parameters; (3) prescription interval of cefoperazone/sulbactam of >7 days or taking other antibiotics simultaneously; (4) received heparin, low-molecular-weight heparin, warfarin or other anticoagulants; or (5) incomplete clinical records. The Naranjo ADR Probability Scale (Naranjo Scale) is used to evaluate the causal relationships between drugs and unexpected clinical events during their use [19]. The conventional total score categories for ADR were as follows: definite, ≥9; probable, 5–8; possible, 1–4; doubtful, ≤0. Patients with scores ≥1 were considered to have cefoperazone/sulbactam-induced hypoprothrombinaemia. Finally, the enrolled patients were randomized at a 7:3 ratio into training and internal validation groups, which were used to establish a nomogram prediction model of cefoperazone/sulbactam-induced hypoprothrombinaemia and to cross-verify the efficacy of the model, respectively. According to the same standards, clinical data of hospitalized patients using cefoperazone/sulbactam at the Third Affiliated Hospital of Xi'an Medical University from January 1, 2023 to June 30, 2023 were collected as the external validation group to further validate the predictive accuracy of the nomogram model.

### Information collection and definitions

Patient information was extracted from the HIS through the CHPS. Because the data was desensitized, visitors couldn't identify information about individual participants during or after data collection. The following patient characteristics were recorded: age, sex, surgery, cancer, bleeding history, length of hospital stay, comorbidities (hypertension, diabetes, cardiovascular disease and chronic kidney diseases), infection site (respiratory tract, intra-abdominal, urinary system, bloodstream, skin and soft tissue, bone and joint, pelvic cavity and intracranial), nutrition status (score of $\geq 3$ on the Nutrition Risk Screening 2002 [20] scale were defined as nutritional risk, and one of $<3$ as no nutritional risk). Laboratory data included platelet (PLT) count (defined as a PLT count $< 50 \times 10^9$/L is considered thrombocytopenia, which can increase the risk of bleeding), hepatic dysfunction (defined as a baseline total bilirubin level of $>3.0$ mg/dL [51.3 μmol/L] or aspartate aminotransferase level of $>250$ U/L), renal dysfunction (defined as a baseline serum creatinine level of $>3.0$ mg/dL [270 μmol/L] or serum urea nitrogen level of $>80$ mg/dL [28.6 mmol/L])]) and haemoglobin. The exposure of patients to antibiotics included the daily dose, frequency, treatment course and cumulative defined daily doses (DDDs) (defined as cumulative doses/DDD, the dose is expressed as the cefoperazone dosage) of cefoperazone/sulbactam during the entire hospitalization.

### Statistical analysis

SPSS software (version 24.0, SPSS, IBM, United States) and R software (version 4.0.3, the R Core Team, United States) were used to conduct the data analysis and model development. All variables of the baseline characteristics did not conform to a normal distribution. Median (range) and number (percentage) values were calculated using descriptive statistics for the baseline characteristics of the participants. Comparisons between groups were performed using the Mann-Whitney $U$ test for continuous variables and chi-square or Fisher's exact tests for categorical variables. Before conducting the analysis, it was confirmed that the assumed preconditions for logistic regression had been met. Univariate logistic regression analysis was used to screen the predictors in the training group. The predictors with $p<0.20$ in the univariate analysis were considered significant and were included in the multivariate analysis. Odds ratios (ORs) and their 95% confidence intervals (CIs) were calculated for the predictors in the univariate and multivariate analyses. The nomogram was constructed based on the independent predictors identified in the multivariate logistics regression using the rms package in R software. The cut-off value was determined using receiver operating characteristic (ROC) curve analysis. The concordance index (c-index) and Hosmer-Lemeshow test were used to evaluate the discrimination and calibration of the prediction model. The c-index can be expressed as the area under the curve (AUC) of the ROC. AUC $>0.7$ indicated that the model had good prediction performance. The prediction model was considered to have acceptable goodness of fit when $p>0.05$ in the Hosmer-Lemeshow text. A two-sided $p<0.05$ was considered significant.

## Results

### Baseline characteristics of the patients

Among patients admitted between 1 January 2020 and 31 December 2022, 8364 patients were treated using cefoperazone/sulbactam at Xi'an Central Hospital, 7471 (89.32%) did not conform with the inclusion criteria, and 893 (10.68%) were included in this study. Among these patients, 378 warning signs were extracted through the CHPS and 283 patients with no causal

relationship with cefoperazone/sulbactam were excluded after independent review by two clinical pharmacists using the Naranjo Scale, and 610 were finally included in the analysis. We determined that 95 of the hospitalized adult patients (95/893, 10.64%) included in our study had cefoperazone/sulbactam-induced hypoprothrombinaemia. This study eventually included 610 patients in the analysis, who were randomized at a 7:3 ratio into the training (*n* = 427) and validation (*n* = 183) groups. A flowchart of patient collection is shown in **Fig 1**. The training

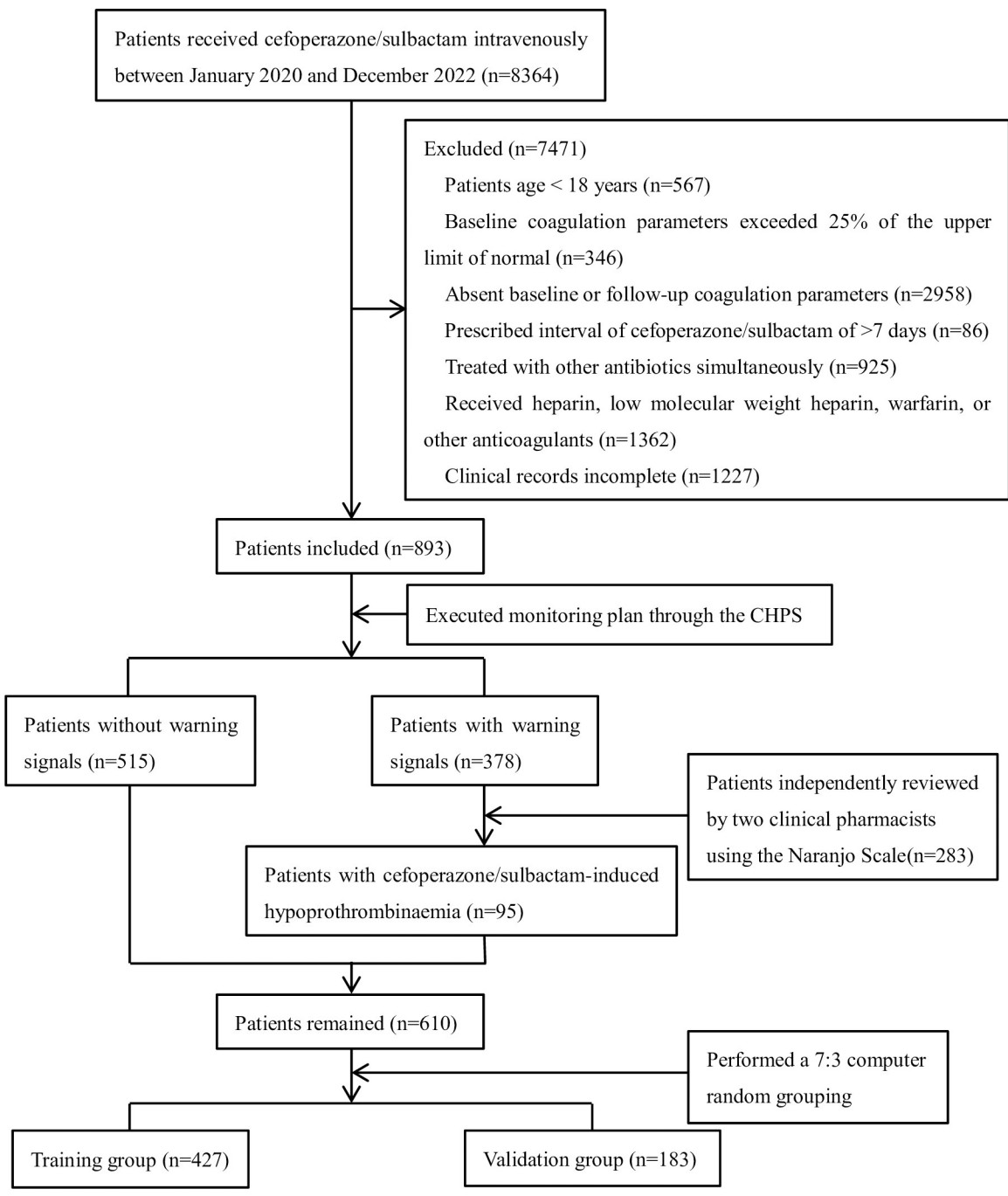

**Fig 1. The flowchart of patient collection.** CHPS indicates Chinese hospital pharmacovigilance system.

group conducted logistic regression analysis and screened independent predictors to establish nomogram model. There were 67 patients (15.69%) in the training group and 28 (15.30%) in the internal validation group who developed hypoprothrombinaemia due to cefoperazone/sulbactam use. The baseline characteristics of the training and internal validation groups are listed in **Table 1**.

## Independent predictors

Univariate and multivariate logistic regression analyses were performed to identify the independent predictors of cefoperazone/sulbactam-induced hypoprothrombinaemia the training group. The univariate analysis indicated that the potential predictors of cefoperazone/sulbactam-induced hypoprothrombinaemia were surgery (yes or no), bleeding history (yes or no), length of hospital stay (continuous days), baseline PLT count $\leq 50 \times 10^9$/L (yes or no), baseline hepatic dysfunction (yes or no), baseline renal dysfunction (yes or no), treatment course (continuous days), cumulative DDDs of cefoperazone/sulbactam and nutritional risk (yes or no). Surgery (OR = 5.279, 95% CI = 2.597–10.729, $p < 0.001$), baseline PLT count $\leq 50 \times 10^9$/L (OR = 2.492, 95% CI = 1.110–5.593, $p = 0.027$), baseline hepatic dysfunction (OR = 12.362, 95% CI = 3.277–46.635, $p < 0.001$), cumulative DDDs (OR = 1.162, 95% CI = 1.162–1.221, $p < 0.001$) and nutritional risk (OR = 16.973, 95% CI = 7.339–39.254, $p < 0.001$) were independently associated with cefoperazone/sulbactam-induced hypoprothrombinaemia in the multivariate analysis. **Table 2** lists the results of the univariate and multivariate logistic regression analyses for the 427 patients in the training group.

## Nomogram development and internal validation

The probability of cefoperazone/sulbactam-induced hypoprothrombinaemia was assessed according to the results of the multivariate logistic regression analysis. The logistic model equation was as follows: ln $(p/1 - p)$ = 1.664×surgery + 0.913×baseline PLT count $\leq 50 \times 10^9$/L + 2.515×baseline hepatic dysfunction + 0.150×cumulative DDDs + 2.832×nutritional risk. The nomogram (**Fig 2**) was based on the proportional conversion of each regression coefficient in the multivariate logistic regression lying ranging from 0 to 100 points. Each variable result was summed to obtain the total score, which was converted to risk-prediction probabilities. The performance of the prediction model was measured using ROC curves and calibration plots with 1000 bootstrap samples to reduce overfitting bias. The ROC curves of the training and internal validation groups are shown in **Fig 3A and 3B**. The nomogram demonstrated good accuracy in predicting the risk of cefoperazone/sulbactam-induced hypoprothrombinaemia, with AUCs of 0.909 (95% CI = 0.875–0.943) and 0.888 (95% CI = 0.832–0.944) for the training and internal validation groups, respectively. The optimal cut-off value for the nomogram-predicted probability was 0.158, which yielded a sensitivity, specificity, and accuracy for estimating the cefoperazone/sulbactam-induced hypoprothrombinaemia risk of 85.1%, 81.4% and 82.0% in the training group, and 75.0%, 89.0% and 86.9% in the internal validation group, respectively. The risk predictions of the nomogram showed good consistency with the actual observed results, and the Hosmer-Lemeshow test yielded $p = 0.475$ (**Fig 4A**) and $p = 0.742$(**Fig 4B**) for the training and internal validation groups, respectively.

## Nomogram external validation

In the external validation group (n = 221), a total of 21 patients experienced hypoprothrombinaemia caused by cefoperazone/sulbactam, with an AUC of 0.837 (95%CI = 0.736–0.938), indicating that the predictive ability of the nomogram for cefoperazone/sulbactam-induced hypoprothrombinaemia in other centres is equally robust (**Fig 3C**). Compared with the

**Table 1. Baseline characteristics in the training and internal validation groups.**

| Variables | Training Group(n = 427) | | | Internal Validation Group(n = 183) | | |
|---|---|---|---|---|---|---|
| | Hypoprothrombinaemia (n = 67) | No hypoprothrombinaemia (n = 360) | *p* value | Hypoprothrombinaemia (n = 28) | No hypoprothrombinaemia (n = 155) | *p* value |
| **Demographic data** | | | | | | |
| Males, n (%) | 39(58.2) | 206(57.2) | 0.894 | 19(67.9) | 88(56.8) | 0.304 |
| Median age, years (IQR) | 48(44–58) | 49(41–58) | 0.868 | 50.5(32.0–58.0) | 49(35–58) | 0.725 |
| Surgery, n (%) | 42(62.7) | 113(31.4) | <0.001 | 16(57.1) | 52(33.5) | 0.021 |
| Cancer, n (%) | 20(29.9) | 82(22.8) | 0.215 | 12(42.9) | 37(23.9) | 0.061 |
| Bleeding history, n (%) | 5(7.5) | 2(0.6) | 0.001 | 0 | 1(0.6) | 1.000 |
| Median length of hospital stay, days (IQR) | 17(13–21) | 16(11–21) | 0.147 | 19(15–26) | 17(12–21) | 0.026 |
| **Comorbidities** | | | | | | |
| Hypertension, n (%) | 15(22.4) | 86(23.9) | 0.876 | 4(14.3) | 36(23.2) | 0.455 |
| Diabetes, n (%) | 14(20.9) | 71(19.7) | 0.868 | 5(17.9) | 29(18.7) | 1.000 |
| Cardiovascular disease, n (%) | 14(20.9) | 62(17.2) | 0.488 | 4(14.3) | 14(9.0) | 0.486 |
| Chronic kidney diseases, n (%) | 6(9.0) | 26(7.2) | 0.614 | 4(14.3) | 7(4.5) | 0.068 |
| **Infection site** | | | | | | |
| Respiratory tract, n (%) | 47(70.1) | 233(64.7) | 0.484 | 19(67.9) | 93(60.0) | 0.529 |
| Intra-abdominal, n (%) | 12(17.9) | 70(19.4) | 0.867 | 9(32.1) | 28(18.1) | 0.122 |
| Urinary system, n (%) | 8(11.9) | 29(8.1) | 0.342 | 2(7.1) | 24(15.5) | 0.378 |
| Bloodstream, n (%) | 3(4.5) | 9(2.5) | 0.412 | 3(10.7) | 1(0.6) | 0.012 |
| Skin and soft tissue, n (%) | 3(4.5) | 12(3.3) | 0.715 | 2(7.1) | 2(1.3) | 0.111 |
| Bone and joint, n (%) | 1(1.5) | 19(5.3) | 0.339 | 2(7.1) | 9(5.8) | 0.677 |
| Pelvic cavity, n (%) | 1(1.5) | 17(4.7) | 0.330 | 0 | 7(4.5) | 0.597 |
| Intracranial, n (%) | 0 | 10(2.8) | 0.374 | 0 | 7(4.5) | 0.597 |
| **Laboratory measurements** | | | | | | |
| Baseline PLT count≤50×$10^9$/L, n (%) | 19(28.4) | 50(13.9) | 0.006 | 6(21.4) | 25(16.1) | 0.583 |
| Baseline hepatic dysfunction, n (%) | 10(14.9) | 12(3.3) | 0.001 | 8(28.6) | 9(5.8) | 0.001 |
| Baseline renal dysfunction, n (%) | 7(10.4) | 19(5.3) | 0.157 | 1(3.6) | 6(3.9) | 1.000 |
| Median PLT count, ×$10^9$/L (IQR) | 116(48–212) | 140(87–187) | 0.374 | 132(45–225) | 148(87–217) | 0.409 |
| Median HB, g/L (IQR) | 106(89–124) | 107(88–121) | 0.947 | 93(87–109) | 113(89–127) | 0.029 |
| **Antibiotics exposure** | | | | | | |
| Median treatment course, days (IQR) | 12(8–15) | 9(6–14) | 0.002 | 12(8–15) | 10(6–14) | 0.055 |
| Median cumulative DDDs (IQR) | 13.5(10.0–19.5) | 8.4(5.1–13.0) | <0.001 | 13.5(9.2–17.8) | 9.0(6.0–14.0) | <0.001 |
| Median nutritional risk, n (%) | 54(80.6) | 110(30.6) | <0.001 | 21(75.0) | 43(27.7) | <0.001 |

Abbreviations: IQR, interquartile range; PLT, platelet; HB, hemoglobin; DDDs, defined daily doses.

**Table 2. Univariate and multivariate logistic regression analyses of predictors for cefoperazone/sulbactam-induced hypoprothrombinaemia.**

| Variables | Univariate | | Multivariate | |
|---|---|---|---|---|
| | Odds ratio (95% CI) | p value | Odds ratio (95% CI) | p value |
| Surgery | 3.672(2.134–6.319) | <0.001 | 5.279(2.597–10.729) | <0.001 |
| Bleeding history | 14.435(2.740–74.060) | 0.002 | | |
| Length of hospital stay | 1.017(0.994–1.040) | 0.142 | | |
| Baseline PLT count≤50×10$^9$/L | 2.454(1.334–4.514) | 0.004 | 2.492(1.110–5.593) | 0.027 |
| Baseline hepatic dysfunction | 5.088(2.100–12.324) | <0.001 | 12.362(3.277–46.635) | <0.001 |
| Baseline renal dysfunction | 2.094(0.844–5.197) | 0.111 | | |
| Treatment course | 1.076(1.034–1.120) | <0.001 | | |
| Cumulative DDDs | 1.127(1.082–1.174) | <0.001 | 1.162(1.105–1.221) | <0.001 |
| Nutritional risk | 9.441(4.950–18.006) | <0.001 | 16.973(7.339–39.254) | <0.001 |

Abbreviations: CI, confidence interval; PLT, platelet; DDDs, defined daily doses.

training and internal validation groups, the correction curves (Fig 4C) showed that the predicted probability of the nomogram model was similar to the actual observation probability ($p = 0.384$), indicating that the nomogram model had good extrapolation performance.

## Discussion

The main findings of our study were as follows: (1) the hypoprothrombinaemia incidence rate among patients who received cefoperazone/sulbactam treatment was not low (10.64%); (2) surgery, baseline PLT count ≤50×10$^9$/L, baseline hepatic dysfunction, cumulative DDDs and nutritional risk were independent predictors of cefoperazone/sulbactam-induced hypoprothrombinaemia; and (3) our nomogram model is feasible for predicting cefoperazone/sulbactam-induced hypoprothrombinaemia risk in hospitalized adult patients.

Several factors were associated with an increased risk of cefoperazone/sulbactam-induced hypoprothrombinaemia. The common mechanisms of cefoperazone/sulbactam-induced hypoprothrombinaemia are inhibiting the activity of vitamin-K-dependent coagulation factors II, VII, IX and X, and interfering with vitamin K metabolism through the N-methylthio-tetrazole (NMTT) side chain of cefoperazone to induce vitamin K deficiency [21], which results in

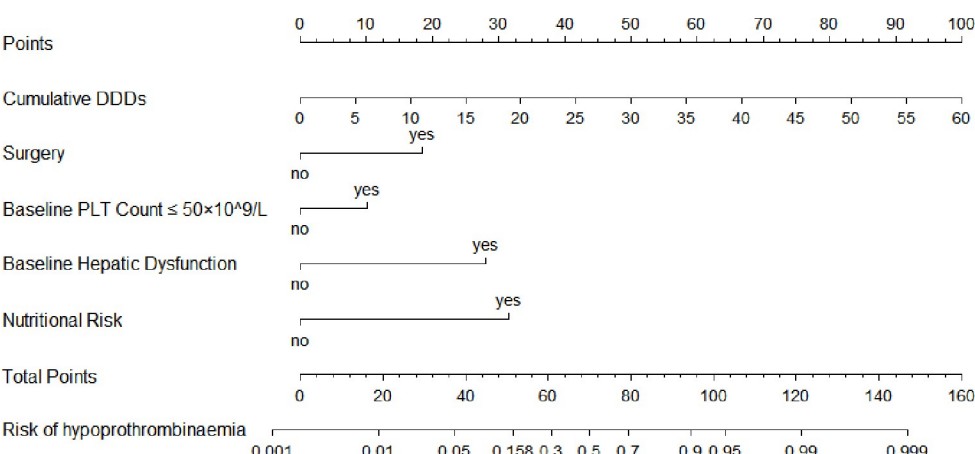

**Fig 2. Nomogram for the prediction of cefoperazone/sulbactam-induced hypoprothrombinaemia risk.** DDDs indicate defined daily doses; PLT indicates platelet.

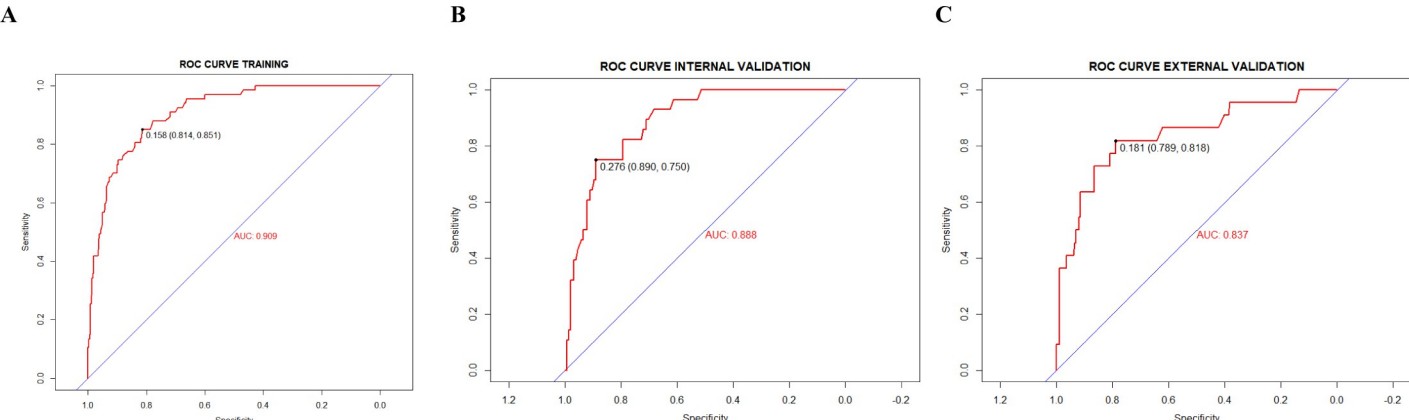

**Fig 3.** The ROC curves in the training (A), internal validation (B) and external validation (C) groups. The AUCs for the training (A), internal validation (B) and external validation (C) groups were 0.909 (95% CI = 0.875–0.943), 0.888 (95% CI = 0.832–0.944) and 0.837 (95% CI = 0.736–0.938), respectively. ROC indicates receiver operating characteristic; AUC indicates area under the curve.

prolonged PT and increased bleeding risk [22, 23]. However, patients with sufficient vitamin K can offset the effect of the NMTT concentration [24]. Unlike anticoagulants, no antibacterial agent has sufficient potential to induce hypoprothrombinaemia in healthy people [25]. There is no significant bleeding risk from antibacterial agents with or without the NMTT side chain in patients at a low risk of hypoprothrombinaemia. This implies that clinical risk factors are important for hypoprothrombinaemia occurrence.

The risk-prediction nomogram for cefoperazone/sulbactam-induced hypoprothrombinaemia was developed using the training group and validated using the validation group. The AUCs for the training, internal validation and external validation groups in this nomogram were 0.909 (95% CI = 0.875–0.943), 0.888 (95% CI = 0.832–0.944) and 0.837 (95%CI = 0.736–0.938), respectively. The Hosmer-Lemeshow tests yielded $p>0.05$ in the training, internal validation and external validation groups. These results demonstrate that the nomogram has a good ability to distinguish between patients with and without cefoperazone/sulbactam-induced hypoprothrombinaemia, without overestimating or underestimating the risk of occurrence. The nomogram model predicts the risk of cefoperazone/sulbactam-induced

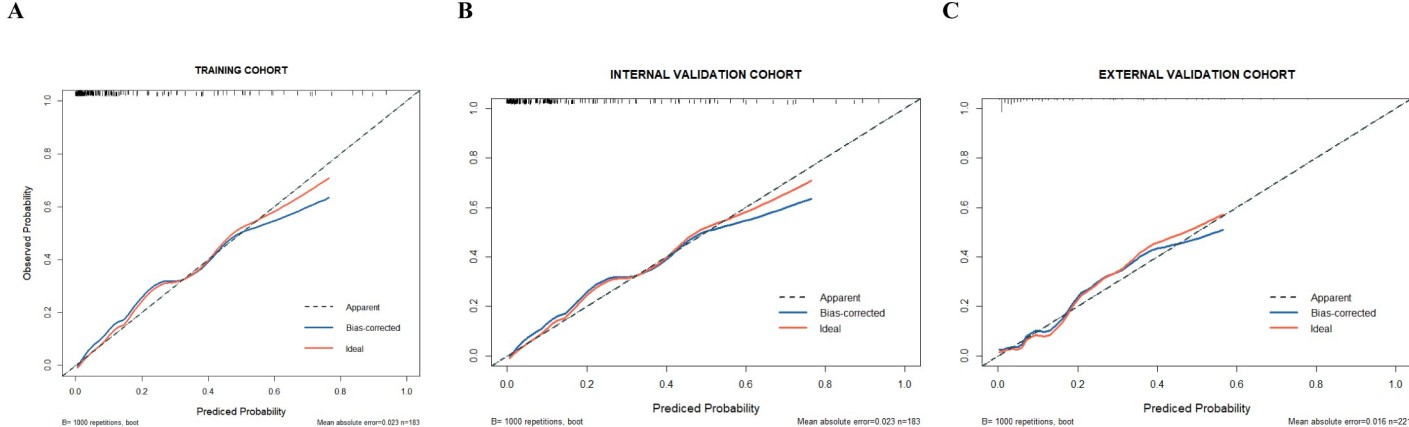

**Fig 4.** The Hosmer-Lemeshow tests in the training (A), internal validation (B) and external validation (C) groups. The x-axis shows the predicted probability of cefoperazone/sulbactam-induced hypoprothrombinaemia, and the y-axis shows the observed probability of cefoperazone/sulbactam-induced hypoprothrombinaemia.

hypoprothrombinaemia in hospitalized adult patients based on five predictors: surgery, baseline PLT count $\leq 50 \times 10^9$/L, baseline hepatic dysfunction, cumulative DDDs and nutritional risk. Previous studies [26, 27] found that anticoagulant use, liver and renal failure, poor nutritional status and high-dose cefoperazone/sulbactam were independent covariates of cefoperazone/sulbactam-induced hypoprothrombinaemia. However, we excluded patients who took anticoagulants during hospitalization to avoid the residual effects of the event of hypoprothrombinaemia. Renal dysfunction was not included as a predictor in the model, which may be attributed to cefoperazone being excreted primarily via bile, and the concentrations of plasma and bile lack the influence of kidney disease. The cefoperazone/sulbactam dose can be down-regulated in patients with renal dysfunction to reduce the hypoprothrombinaemia risk [26]. Nevertheless, the drug clearance has been found to be significantly decreased and the half-life prolonged in patients with hepatic dysfunction [28]. Patients with severe hepatic dysfunction are at increased risk of developing hypoprothrombinaemia due to their impaired ability to synthesize coagulation factors [29]. Concurrent cefoperazone/sulbactam administration could further hinder hepatic synthesis of coagulation factors.

We found that surgery and nutritional risk contributed approximately 18 and 32 points to the predicted total score of the nomogram, respectively. Surgery was the most common risk factor for cefoperazone/sulbactam-induced hypoprothrombinaemia [7, 30]. This was attributed to decreased dietary intake and impaired gastrointestinal function during the postoperative period that resulted in reduced vitamin K absorption. We also found that nutritional risk was significantly associated with increased hypoprothrombinaemia risk (OR = 16.973, 95% CI = 7.339–39.254). At least one risk factors for poor nutritional status was present in almost all cases of β-lactam-antimicrobial-associated hypoprothrombinaemia [27]. Patients with poor nutritional status had insufficient vitamin K intake, which might increase the risk of cefoperazone/sulbactam-induced hypoprothrombinaemia. Furthermore, while some researchers [26, 31] have observed an effect from cefoperazone/sulbactam on PLT function, few studies have explored the relationship between PLT dysfunction and cefoperazone/sulbactam-induced hypoprothrombinaemia. Our study concluded that a baseline PLT count of $\leq 50 \times 10^9$/L has predictive significance for cefoperazone/sulbactam-induced hypoprothrombinaemia (OR = 2.492, 95% CI = 1.110–5.593). Nevertheless, its mechanism of action remains unknown. This may be related to the decreased prothrombin activity on the surface of PLTs caused by lower PLT count [32]. PLT count also has a synergistic effect with cefoperazone/sulbactam resulting in increasing hypoprothrombinaemia risk.

A study by Strom et al. [27] suggested that there was a significant dose–response relationship between cefoperazone and hypoprothrombinaemia, and that a cefoperazone dose of >4.5 g/day markedly increased the risk of bleeding. A nationwide nested case–control study [27] found that the use of hypoprothrombinaemia-inducing cefoperazone was associated with an increased risk of bleeding, which was significantly higher in patients with >5 cumulative DDDs than those with <3. However, we used DDD to calculate the cumulative DDDs of cefoperazone/sulbactam, considering the effects of daily dose and course of cefoperazone/sulbactam treatment on hypoprothrombinaemia. Our study found that patients who received cefoperazone at a daily dose of >4g and with a treatment course of >14 days may have a higher risk of hypoprothrombinaemia, and thus we could speculate that higher cumulative DDDs would mean higher hypoprothrombinaemia risk. The role of cumulative DDDs in the prediction of cefoperazone/sulbactam-induced hypoprothrombinaemia should be assessed clinically.

The final prediction model did not include bleeding history, length of hospital stay, baseline renal dysfunction or cefoperazone/sulbactam treatment course. These factors were significantly correlated in the univariate analysis, but the interference of confounding factors needed to be removed before adding putative factors to our model. Moreover, the final predictive

nomogram is relatively simple and feasible to apply in clinical practice. According to the clinical characteristics of each patient, the corresponding points of each factor were obtained by referring to the nomogram. The points of the five factors are added together to derive total points, which are converted to predicted probabilities. The cut-off value of the model is 0.158. Patients with the prediction probability of 0.158 or more are a high-risk of cefoperazone/sulbactam-induced hypoprothrombinaemia.

Early risk stratification of patients during hospitalization is important for their clinical management. Assessing the risk of cefoperazone/sulbactam-induced hypoprothrombinaemia can enable clinicians to individually adjust the intensity of laboratory monitoring, and increase vigilance in the presence of nutritional risk and high-dose administration of cefoperazone/sulbactam. It is still controversial whether patients who receive cefoperazone/sulbactam should also receive vitamin K to prevent hypoprothrombinaemia occurrence [10, 28]. In our study, patients identified as high risk through early screening were recommended to receive vitamin K to prevent hypoprothrombinaemia occurrence. In the event of cefoperazone/sulbactam-induced hypoprothrombinaemia, vitamin K or fresh frozen plasma should be administered promptly in order to reverse the condition.

One strength of our study was that it was the first to analyse cefoperazone/sulbactam-induced hypoprothrombinaemia episodes in hospitalized patients through the CHPS. Another was the construction of a risk-prediction nomogram for cefoperazone/sulbactam-induced hypoprothrombinaemia, which will allow physicians to calculate total scores and assess hypoprothrombinaemia risk before the intervention and to take further preventive measures to reduce hypoprothrombinaemia occurrence.

There were some limitations in our study. First, the analysis had a retrospective design, and so it was information bias in data acquisition. Second, nearly one-third of patients treated using cefoperazone/sulbactam lacked baseline or follow-up coagulation parameters and were not included in this study, which may have biased the sample selection. Third, because only the changes in the coagulation parameters PT and APTT were used to diagnose hypoprothrombinaemia, it is likely that some positive cases were missed and the overall incidence was underestimated. Finally, a large prospective study is required to further confirm the predictive performance of the nomogram.

## Conclusions

We have developed and validated a nomogram based on five predictors to predict cefoperazone/sulbactam-induced hypoprothrombinaemia occurrence in hospitalized adult patients. The use of this prediction model can allow closer monitoring and early treatment to help prevent hypoprothrombinaemia occurrence in high-risk patients. Further research is needed to evaluate the applicability of the prediction model.

## Supporting information

**S1 File. The patient characteristics.**
(XLSX)

**S2 File. Nomogram record.**
(PDF)

**S3 File. External validation record.**
(PDF)

## Author Contributions

**Conceptualization:** Hehe Bai, Xiaonian Han, Jinping Wang, Lirong Peng.

**Data curation:** Hehe Bai, Xiaojing Nie, Xiaonian Han, Jinping Wang, Lirong Peng.

**Formal analysis:** Hehe Bai, Lirong Peng.

**Funding acquisition:** Hehe Bai, Lirong Peng.

**Investigation:** Hehe Bai, Xiaojing Nie, Yanqin Yao, Xiaonian Han, Jinping Wang.

**Methodology:** Huan Li, Xiaojing Nie, Xiaonian Han, Lirong Peng.

**Project administration:** Hehe Bai, Xiaojing Nie, Lirong Peng.

**Resources:** Hehe Bai, Jinping Wang.

**Software:** Huan Li.

**Supervision:** Xiaojing Nie, Xiaonian Han.

**Validation:** Huan Li, Yanqin Yao.

**Visualization:** Yanqin Yao, Jinping Wang.

**Writing – original draft:** Hehe Bai.

**Writing – review & editing:** Huan Li, Lirong Peng.

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
