## [Decision Letter · Decision Letter 0]

11 Aug 2023

PONE-D-23-21781Development and Validation of a Nomogram for Predicting Cefoperazone/Sulbactam-Induced Hypoprothrombinaemia in Hospitalized Adult PatientsPLOS ONE

Dear Dr. PENG,

Thank you for submitting your manuscript to PLOS ONE. After careful consideration, we feel that it has merit but does not fully meet PLOS ONE’s publication criteria as it currently stands. Therefore, we invite you to submit a revised version of the manuscript that addresses the points raised during the review process.

We look forward to receiving your revised manuscript.

Kind regards,

Sairah Hafeez Kamran, PhD

Academic Editor

PLOS ONE

Journal Requirements:

"This work was supported by the Xi'an Science and Technology Program (No.22YXYJ0015) and the Scientific Research Foundation of Xi 'an Central Hospital (No.2022YB06)."

6. We note you have included a table to which you do not refer in the text of your manuscript. Please ensure that you refer to Tables 1 and 2 in your text; if accepted, production will need this reference to link the reader to the Table.

Reviewers' comments:

Reviewer's Responses to Questions

**Comments to the Author**

1. Is the manuscript technically sound, and do the data support the conclusions?

Reviewer #1: Yes

Reviewer #2: Yes

2. Has the statistical analysis been performed appropriately and rigorously? 

Reviewer #1: Yes

Reviewer #2: Yes

3. Have the authors made all data underlying the findings in their manuscript fully available?

Reviewer #1: Yes

Reviewer #2: Yes

4. Is the manuscript presented in an intelligible fashion and written in standard English?

Reviewer #1: Yes

Reviewer #2: Yes

5. Review Comments to the Author

Reviewer #1: Thank you for this opportunity to review this interesting study “Development and Validation of a Nomogram for Predicting Cefoperazone/Sulbactam-induced Hypoprothrombinaemia in Hospitalized Adult Patients”. The study addresses an important clinical issue related to cefoperazone/sulbactam-induced hypoprothrombinaemia, which is associated with longer hospital stays and increased risk of death. Identifying patients at risk early can help in preventing the adverse outcome and timely intervention. The study included a substantial number of hospitalized patients and then divided them into training and validation groups, which enhances the generalizability of the findings. The use of multivariate logistic regression to identify independent predictors of cefoperazone/sulbactam-induced hypoprothrombinaemia and the development of a nomogram can potentially provide a practical tool for clinicians to predict the risk in individual patients. The area under the receiver operating characteristic curves (AUC) for both the training and validation groups were relatively high (0.909 and 0.888, respectively), indicating good predictive performance of the model. Nevertheless, the study follows a retrospective cohort design, which can be prone to bias and limitations related to data collection and confounding factors. Prospective studies could provide stronger evidence. Being a single-center study conducted in Xi’an Central Hospital, the generalizability of the results to other healthcare settings and patient populations may be limited. Validation in multiple centers would strengthen the findings. The study enrolled patients treated with cefoperazone/sulbactam at Xi’an Central Hospital only, which could introduce selection bias as the patient population may not be representative of all hospitalized patients. The study's reliance on data extracted from the hospital information system (HIS) may introduce inaccuracies or missing data, affecting the reliability of the results. While the nomogram showed good performance within the study population, it would be beneficial to validate the model externally in different patient cohorts to assess its general applicability. Authors should address these limitations of study. Overall, the paper is well written with good collection of data; however, authors are requested to collect and add more information from the papers/literature that are published in 2020-2023. The author should remove old references (6,7,11 etc) because article lacks in this area.

Reviewer #2: The study is interesting and discussing an important topic in the domain of health services. However, I have concern about some points as below.

Method

Study design and collection

Please, further explain the following sentence “Written informed consent was obtained from the individuals for the publication of any potentially identifiable images or data included in this article.” I mean, please explain how the informed consent was obtained from the individuals.

Authors mentioned “The conventional total score categories for ADR were as follows: definite, ≥ 9; probable, 5 – 8; possible, 1 – 4; doubtful, ≤ 0”. Is it possible for a score to be < 0?

Statistical Analysis

Before conducting the analysis there is a need to assure that basic assumptions for logistic regression were met.

Results

Baseline Characteristics of the Patients

The authors mentioned “893 (10.68%) were finally included in this study”, and again “This study eventually included 610 patients in the analysis”. Why only 610, out of the 893 included in this study, were analyzed? This is confusing.

Also, the authors mentioned that “We finally determined that 95 of the hospitalized adult patients (95/893, 10.64%) included in our study had cefoperazone/sulbactam-induced hypoprothrombinaemia”. I’m confused, what is the exact number of patients included in the logistic regression analysis?

Discussion

The two paragraphs from “We believe that hypoprothrombinaemia is a frequent adverse event in hospitalized patients ...” till “Our study therefore aimed to develop a simple and easy-to-use nomogram based on clinical risk factors to help clinicians more quickly and accurately identify patients with a potential risk of cefoperazone/sulbactam-induced hypoprothrombinaemia.” Better to be moved to the “Introduction” section, and delete the sentence “In our study, the overall incidence rate of cefoperazone/sulbactam-induced hypoprothrombinaemia among inpatients was 10.64%(95/893), which was consistent with that in the literature.”

Please, discuss the unexpected findings (if you think there is any).

The last subtitle I assume it is ‘Conclusion’ not ‘Discussion’.

6. PLOS authors have the option to publish the peer review history of their article (what does this mean?). If published, this will include your full peer review and any attached files.

Reviewer #1: **Yes: **Zikria Saleem

Reviewer #2: **Yes: **Abubakr Abdelraouf Alfadl

---

## [Author Response · Author response to Decision Letter 0]

31 Aug 2023

Academic Editor 

Response1: Our manuscript has been modified according to PLOS ONE style templates. 

Response2: If the funders had no role, as stated in the manuscript: "The funders had no role in study design, data collection and analysis, decision to publish, or preparation of the manuscript." 

Response3: We have changed the data availability statement in the cover letter.

Response4: The corresponding author's ORCID iD is https://orcid.org/0009-0001-9586-3719.

Response5: The ethics statement have been amended ethics statement ethics statement in the Methods section of the manuscript.

Response6: It has been confirmed that only Tables 1 and 2 are referred to for the manuscript.

Response7: Supporting information files have been added at the end of the manuscript.

Response8: Papers that have been retracted are not cited in the manuscript. According to the suggestions of reviewer #1, references [7], [31] and [34] are deleted, references [6], [11], [20], [21] and [25] are replaced, and references [29] are added in the manuscript.

Reviewer #1

Response1: The nomogram model was externally verified in the Third Affiliated Hospital of Xi 'an Medical University, with an AUC of 0.837 (95%CI=0.736–0.938), indicating that the predictive ability of the nomogram for cefoperazone/sulbactam-induced hypoprothrombinaemia in other centres is equally robust. The manuscript has been modified accordingly.

Response2: The HIS can sort out and integrate the data information from laboratory information system (LIS), picture archiving and communication system (PACS) and radiology information system (RIS) to avoid inaccuracies or missing data and ensure data reliability.

Response3: We have updated the references and removed the old ones. References [7], [31] and [34] were deleted from the original manuscript, references [6], [11], [20], [21] and [25] were replaced, and references [29] were added.

Reviewer #2

Study design and collection

This study is retrospective and patient informed consent has been waived in ethical statement. We have amended our statements regarding patient informed consent.

The Naranjo ADR Probability Scale was developed to help standardize assessment of causality for all adverse drug reactions. The Naranjo Scale contains 10 subscales, with Yes, No, and Not known or not done options for each item. Among them, the "No" option of the 2nd and 4th subscales has a score of -1, and the "Yes" option of the 5th and 6th subscales has a score of -1, and there may be cases where the score is less than or equal to 0. View the Naranjo scale of use at https://www.evidencio.com/models/show/661.

Statistical Analysis

Before conducting the analysis, it was confirmed that the assumed preconditions for logistic regression had been met.

Baseline Characteristics of the Patients

Among 893 patients, 378 warning signs were extracted through the CHPS and 283 patients with no causal relationship with cefoperazone/sulbactam were excluded after independent review by two clinical pharmacists using the Naranjo Scale. Because the 283 patients with warning signs could be related to factors other than cefoperazone/sulbactam, they were not included in the final analysis. Finally, 610 patients were included in the analysis. 

Among the 378 patients with warning signs, 283 patients were excluded from possible factors other than cefoperazone/sulbactam, and 95 patients were finally determined to have cefoperazone/sulbactam-induced hypoprothrombinaemia (Fig 1). This study eventually included 610 patients in the analysis, who were randomized at a 7:3 ratio into the training (n=427) and validation (n=183) groups. The training group (n=427) conducted logistic regression analysis and screened independent predictive factors to establish nomogram model.

Discussion

The discussion has been partially moved to the introduction. The discussion has been partially moved to the introduction. The sentence “In our study, the overall incidence rate of cefoperazone/sulbactam-induced hypoprothrombinaemia among inpatients was 10.64%(95/893), which was consistent with that in the literature.” was also deleted.

The last subtitle has been corrected to conclusion.

---

## [Decision Letter · Decision Letter 1]

4 Sep 2023

Development and Validation of a Nomogram for Predicting Cefoperazone/Sulbactam-Induced Hypoprothrombinaemia in Hospitalized Adult Patients

PONE-D-23-21781R1

Dear Dr. PENG,

We’re pleased to inform you that your manuscript has been judged scientifically suitable for publication and will be formally accepted for publication once it meets all outstanding technical requirements.

Kind regards,

Sairah Hafeez Kamran, PhD

Academic Editor

PLOS ONE

Comments of Reviewers

Reviewer #1: All comments have been addressed

Reviewer #2: All comments have been addressed

2. Is the manuscript technically sound, and do the data support the conclusions?

Reviewer #1: Partly

Reviewer #2: (No Response)

3. Has the statistical analysis been performed appropriately and rigorously? 

Reviewer #1: I Don't Know

Reviewer #2: (No Response)

4. Have the authors made all data underlying the findings in their manuscript fully available?

Reviewer #1: Yes

Reviewer #2: (No Response)

5. Is the manuscript presented in an intelligible fashion and written in standard English?

Reviewer #1: Yes

Reviewer #2: (No Response)

6. Review Comments to the Author

Reviewer #1: (No Response)

Reviewer #2: (No Response)

7. PLOS authors have the option to publish the peer review history of their article (what does this mean?). If published, this will include your full peer review and any attached files.

Reviewer #1: No

Reviewer #2: **Yes: **Abubakr Abdelraouf Alfadl

---

## [Editor Report · Acceptance letter]

15 Sep 2023

PONE-D-23-21781R1 

Development and Validation of a Nomogram for Predicting Cefoperazone/Sulbactam-Induced Hypoprothrombinaemia in Hospitalized Adult Patients 

Dear Dr. Peng:

I'm pleased to inform you that your manuscript has been deemed suitable for publication in PLOS ONE. Congratulations! Your manuscript is now with our production department. 

Kind regards, 

on behalf of

Dr. Sairah Hafeez Kamran 

Academic Editor

PLOS ONE